# The Time-Dependent Association of Torque Teno Virus Load with the Level of SARS-CoV-2 S1 IgG Antibodies Following COVID-19 Vaccination in Kidney Transplant Recipients

**DOI:** 10.3390/v15112189

**Published:** 2023-10-31

**Authors:** Céline Imhof, Lianne Messchendorp, Debbie van Baarle, Ron T. Gansevoort, Coretta Van Leer-Buter, Jan-Stephan F. Sanders

**Affiliations:** 1Department of Internal Medicine, Division of Nephrology, University Medical Center Groningen, 9713 GZ Groningen, The Netherlands; 2Department of Medical Microbiology and Infection Prevention, University Medical Center Groningen, 9713 GZ Groningen, The Netherlands

**Keywords:** COVID-19 vaccination, Torque teno virus, kidney transplantation

## Abstract

Kidney transplant recipients (KTR) show an impaired humoral immune response to COVID-19 vaccination due to their immunocompromised status. Torque teno virus (TTV) is a possible marker of immune function. This marker may be helpful in predicting the immune response after COVID-19 vaccination in order to decide which vaccination strategy should be applied. We therefore investigated whether TTV load is associated with the humoral response after COVID-19 vaccination. Of the KTR who participated in two prospective vaccination studies and received two to four doses of the mRNA-1273 COVID-19 vaccine, 122 were included. TTV load was measured prior to vaccination, and S1 IgG antibody levels were measured 28 days after vaccination. TTV load was independently inversely associated with S1 IgG antibodies after COVID-19 vaccination (B: −2.19 (95% CI: −3.6–−0.8), *p* = 0.002). Interestingly, we found a significant interaction between TTV load and time after transplantation (*p* = 0.005). When patients were longer after transplantation, TTV load was less predictive for S1 IgG antibody response after vaccination compared to patients that were shorter after transplantation. Our data suggest that TTV load is a good marker in predicting COVID-19 vaccination antibody response and may be helpful in selecting a strategy shortly after transplantation. However, this marker should be handled with caution longer after transplantation.

## 1. Introduction

Kidney transplant recipients (KTR) show an impaired humoral response following COVID-19 vaccination [1,2,3]. Despite receiving multiple vaccinations, their seroresponse remains significantly lower than healthy individuals [4,5]. It has been demonstrated that immunosuppression due to the use of immunosuppressive agents, especially mycophenolate mofetil or mycophenolic acid (MMF), is an important factor in this reduced seroresponse [6,7]. However, there are significant interindividual differences in seroresponse after vaccination in KTR using MMF. Furthermore, an impaired seroresponse is also found in patients not using MMF. Therefore, identifying a biomarker that reflects the immune system’s status and that can predict the seroresponse to vaccination is imperative. This could aid in guiding vaccination schedules or strategies in this high-risk population.

Torque teno virus (TTV), a non-pathogenic single-stranded DNA virus with a high prevalence in the general population [8], has recently been introduced as a promising candidate for assessing immune responsiveness in KTR. It is used to find the right balance between preventing rejection and susceptibility to infection when dosing immunosuppressive agents [9,10]. TTV is suppressed by a functional immune system in immunocompetent individuals, whereas an elevated TTV load indicates a more immunocompromised state. For that reason, TTV load might reflect the ability to elicit an immune response after COVID-19 vaccination in KTR. Several studies have recently suggested an inverse association between TTV load and seroresponse after COVID-19 vaccination [11,12,13,14]. These studies assessed seroresponse as a dichotomous variable, i.e., being present or absent, using a cut-off value. The seroresponse cut-off was different between these studies. Moreover, these studies attempted to calculate a threshold on TTV load that could predict the seroresponse after COVID-19 vaccination and found different thresholds, which makes these results difficult to implement. Furthermore, we have recently demonstrated that antibody levels after COVID-19 vaccination are associated in a log-linear relationship with the occurrence and severity of COVID-19 [15], i.e., the higher the antibody level, the lower the chance of occurrence and severe COVID-19. It would, therefore, be more plausible and clinically relevant to investigate the humoral response after COVID-19 vaccination as a continuous variable.

Therefore, we aimed to investigate whether Torque teno virus (TTV) load is associated with the humoral response after COVID-19 vaccination expressed on a continuous scale.

## 2. Methods

We included KTR who participated in two vaccination studies that were performed in the UMC Groningen in the framework of the RECOVAC (the REnal patients COVID-19 VACcination) Consortium. The first study was conducted between 1 February and 31 May 2021 (www.ClinicalTrials.gov NCT04741386 (accessed on 1 May 2023)). This study investigated the efficacy and safety of the first two doses of a COVID-19 vaccination in patients with CKD stages 4/5, on dialysis, or alive with a kidney transplant compared to controls. The second study was performed between 20 October 2021, and 5 February 2022, and investigated the immunogenicity of various vaccination strategies following a third or fourth dose of a COVID-19 vaccine in KTR (www.ClinicalTrials.gov NCT05030974 (accessed on 1 September 2021)). Ethical approval for these studies was obtained from the Dutch Central Committee on Research Involving Human Subjects and the central ethics committee at the UMC Groningen (NL76215.042.21 and NL78963.042.21, respectively).

### 2.1. Study Participants

Subjects received two mRNA-1273 COVID-19 vaccinations (Moderna Biotech Spain, S.L.) with an interval of 28 days between the vaccinations according to the manufacturer’s instructions. A subset of participants received a 3rd or 4th COVID-19 vaccination with the mRNA-1273 COVID-19 vaccine. This subset of subjects were previously antibody non-responders or at 28 days after the second or third dose of an mRNA-based COVID-19 vaccine. They were randomly assigned to either continuation of mycophenolate mofetil or mycophenolic acid (mycophenolate mofetil+) or discontinuation of mycophenolate mofetil or mycophenolic acid (mycophenolate mofetil−) from 1 week before until one week after vaccination, as previously published [5]. Non-response was defined as anti-SARS-CoV-2 RBD IgG < 50 binding units per mL (BAU/mL) or as S1-specific IgG antibody concentration < 10 BAU/mL. Since there was no difference in antibody response between the two groups, they are analyzed as one group.

To assess immunogenicity, blood samples were collected at baseline (i.e., before vaccination) and 28 days after the second, third, or fourth vaccination, respectively.

A flowchart of subject enrollment of TTV measurement is depicted in Figure 1. KTR who completed follow-up on day 28 after the vaccination and of whom serum samples were available were included (n = 128). Previous COVID-19-infected patients were excluded. In total, 123 KTR were included and analyzed.

### 2.2. SARS-CoV-2 Spike S1-Specific IgG Antibody Response

SARS-CoV-2 Spike S1-specific IgG antibodies were measured in serum samples by a validated fluorescent bead-based multiplex immunoassay (MIA) with a specificity and sensitivity of 99.7% and 91.6%, respectively, as previously described [16,17]. Concentrations were interpolated from a reference consisting of pooled sera using a 5-parameter logistic fit and NIBSC/WHO COVID-19 reference serum 20/136 and expressed as international binding antibody units per mL (BAU/mL). Participants were classified as responders or non-responders based on seroconversion with a threshold for seropositivity based on receiver operator curve analysis set at S1-specific IgG antibody concentration ≥ 10 BAU/mL [17,18].

### 2.3. Quantitative TTV PCR

DNA extraction from baseline serum samples and amplification of DNA was performed as previously described [13,19]. In brief, DNA extraction was carried out using the eMAG Nucleic Acid Extraction System (BioMerieux, Marcy, France). For DNA amplification and quantification, the Argene R-Gene TTV quantification kit (BioMerieux) was used on an Applied Biosystems 7500 (Thermo Fisher, Waltham, MA, USA) according to the manufacturer’s instructions. The R gene assay is designed to detect the majority of TTV genotypes (1, 6, 7, 8, 10, 12, 15, 16, 19, 27 and 28). Due to limited sample volumes, 100 µL, a 1 in 2 dilution using DMEM was performed prior to sample extraction (ThermoFisher). Results are expressed in log copies/mL. Some samples had an undetectable viral load and of 17 patients, subsequent samples were available and tested from 28 days after vaccination. Two patients had a detectable TTV load. The new detectable load was calculated = (value + 1)/2. The other patients were excluded as we do not know if the patient carries the virus.

### 2.4. Statistical Analysis

Continuous data are presented as mean SD or as median IQR in case of non-normal distribution. Categorical data are presented as percentages. Depending on data type and distribution, differences between groups were tested using an independent sample *t*-test, Mann–Whitney U test, or Pearson χ^2^ test. Differences in more than two categories were tested using one-way ANOVA, Kruskal–Wallis, or Pearson chi-square test, depending on data distribution. First, determinants of TTV load were investigated. After that, the association between all the baseline clinical parameters and baseline TTV load was analyzed univariably using linear regression analysis. Variables with an alpha of <0.1 were subsequently included in the multivariable model, followed by a model using a backward elimination procedure. In the latter, the least significant variables were removed stepwise until none met the criterion of *p*-value ≥ 0.05. Thereafter, predictors of S1 IgG antibody level were investigated similarly by analyzing the association between baseline clinical parameters, TTV load, and S1 IgG antibody levels at 28 days after vaccination. Interactions with S1 IgG antibody levels were tested on all significant associated variables. If an interaction was found, results were analyzed stratified for this specific variable. Time after transplantation was divided between shorter after transplantation (≤24 months) and longer after transplantation (>24 months) because the TTV load reaches a stable phase around 24 months [20,21,22]. All analyses were performed with IBM SPSS statistics version 28.0 (SPSS, Chicago, IL, USA). Figures were created with GraphPad Prism version 9·00 (GraphPad Software, San Diego, CA, USA). A two-sided *p*-value of less than 0.05 was adopted to denote significance and corrected in case of multiple testing using Bonferroni correction unless stated otherwise.

## 3. Results

The selection of the included KTR is depicted in a flowchart in Figure 1. Of the total 128 patients, 122 were included. Of these patients, 28 KTR had undetectable TTV loads (23.0%) and were excluded for further analysis. This left 94 KTR for analysis. Baseline characteristics of these 94 patients are shown in Table 1. The mean age was 58.2 ± 12.2 years, the mean eGFR was 50.9 ± 18.9 mL/min/1.73 m^2^, and the median time after transplantation was 41.0 (13.0–85.0) months. The majority of KTR received two COVID-19 vaccinations (53.2%), and the median S1 IgG antibody level after the last COVID-19 vaccination was 90.0 (5.2–527.8) BAU/mL. Of note, the baseline characteristics of the subjects included in the present analyses and those excluded were similar, except for time after transplantation (41.0 (13.0–85.0) versus 69.5 (40.8–94.8) months, *p* = 0.03, respectively; Appendix A).

### 3.1. Associations between Baseline Characteristics and TTV Load at Baseline

Overall, the median TTV load at baseline was 3.78 (2.89–5.00) log copies/mL. There were no significant differences in TTV load between KTR who either received two versus three or four doses of the vaccine (4.1 (3.1–5.1) vs. 3.6 (2.5–4.7) log copies/mL; *p* = 0.07). When TTV was divided into tertiles (Table 1), only time after transplantation and tacrolimus trough levels were significantly different between the three groups.

Subsequently, we performed a linear regression analysis with TTV load as the independent variable (Table 2). The univariable analysis showed that KTR with higher TTV load was shorter after transplantation, had higher tacrolimus trough levels, and more often hypertension (−0.15 (CI: −0.2–−0.1), *p* < 0.001, (0.61 (CI: 0.3–1.0), *p* < 0.001 and (0.10 (CI: 0.02–0.18), *p* = 0.02; respectively) (Table 2). However, in the multivariable analysis, only time after transplantation and hypertension were significantly associated with TTV load (−0.15 (CI: −0.2–−0.1), *p* < 0.001, and 0.1 (CI: 0.01–0.2), *p* = 0.03).

### 3.2. Association with S1 IgG Antibody Level

We performed a linear regression analysis to study whether, besides TTV load, other baseline characteristics were associated with S1 IgG antibody level after vaccination (Table 3). KTR with higher S1 IgG antibody levels were longer after transplantation, had a higher eGFR, a lower TTV load, lower tacrolimus trough levels, were more likely to have received a kidney from a living donor and to have received a third or fourth COVID-19 vaccination (0.62 (CI: 0.1–1.1), *p* = 0.02; 0.03 (CI: 0.1–0.4), *p* < 0.001; −2.30 (CI: −3.8–−0.8), *p* = 0.004; −3.7 (CI: −6.3–−1.2), *p* = 0.004; −0.60 (CI: −1.2–−0.02), *p* = 0.04 and 0.60 (CI: 0.04–1.1), *p* = 0.04). After performing a stepwise backward analysis, only TTV load, eGFR, mycophenolate mofetil use, and the number of received COVID-19 vaccinations remained significantly associated. Subsequently, we tested the final model for interactions and found a significant interaction between time after transplantation and TTV load (4.01 (CI: 0.8–7.2), *p* = 0.01) in predicting antibody response to vaccination, implying that TTV load longer after transplantation was less strongly associated with S1 IgG antibody levels. Given this result, we divided the cohort in KTR who were shorter (n = 32) and longer than 24 months after transplantation (n = 62), based on the literature that states that TTV load reaches a stable phase around 24 months after transplantation. We studied the association between the TTV load and S1 IgG antibody level after vaccination separately in these groups (Figure 2 and Table 4). Only in the first 24 months after transplantation, there was a significant association between TTV load at baseline and S1 IgG antibody level at 28 days after vaccination (−5.3 (CI: −8.3–−2.3), *p* = 0.01), whereas, in patients longer than 24 months after transplantation, such association was absent (−0.61 (CI: −2.7–1.5), *p* = 0.55). In contrast, eGFR, donor type, and number of vaccinations remained significantly associated with S1 IgG antibody levels in the subgroup later after transplantation (0.03 (CI: 0.2–0.5), *p* < 0.001; −0.62 (CI: −1.2–−0.03), *p* = 0.04; and 0.57 (CI: 0.03–1.1), *p* = 0.04, respectively).

## 4. Discussion

This study shows an inverse association between the TTV load and the level of SARS-CoV-2 S1 IgG antibodies following COVID-19 vaccination in KTR. In addition, TTV load correlated with time after transplantation, resulting in a strong inverse association between TTV load and S1 IgG antibody levels after COVID-19 vaccination in KTR only shortly after transplantation.

These results are in accordance with most literature on the association between TTV load and humoral response [11,12,13,14]. However, previous studies have primarily focused on analyzing seroresponse as a dichotomous variable rather than a continuous variable, as was completed in the present study. Currently, the most common SARS-CoV-2 virus variant is the Omicron EG.5 strain, which contains numerous mutations compared to the original Wuhan type [23]. With new SARS-CoV-2 variants, higher levels of antibodies are required to reach adequate neutralizing antibodies for protection against COVID-19. Therefore, conclusions based on analyzing a fixed cut-off value for the assessment of seroresponse can be outdated. In addition, there appears to be a log-linear association between antibody levels following COVID-19 vaccination and the occurrence and severity of COVID-19 in KTR [15]. Consequently, striving for as high an antibody level as possible rather than an antibody level above a cut-off value seems preferable in KTR.

Previous studies investigating TTV load as a predictor of the humoral response after COVID-19 vaccination often controlled for the effect of time after transplantation but did not find a significant association of time after transplantation in multivariable analyses [24]. Causes might have been the limited power of these smaller studies or the fact that seroresponse was studied as a dichotomous instead of a continuous variable, leading to a loss of power. When studying the humoral response after vaccination as a continuous variable, we demonstrated that TTV was highly predictive of the humoral response shortly after kidney transplantation, whereas this association was lost longer after transplantation. Furthermore, there was an association between tacrolimus trough levels and TTV load. This could also be related to time after transplantation, as longer after transplantation, in general, lower tacrolimus trough levels are pursued. However, we did not find a significant interaction between tacrolimus trough levels and time after transplantation in predicting antibody levels after vaccination. Of note, a relationship between tacrolimus trough level and TTV load has been described before [22]. Moreover, almost 25% of our cohort had an undetectable TTV load, of whom it was impossible to predict the vaccination response. Therefore, together with the effect of time after transplantation, using TTV load to guide vaccination schedules or strategies is not straightforward and may only be applicable for some KTR. In case a high TTV load is found in those KTRs, repeat vaccination should be considered in that individual or an alternative vaccination strategy, for example, applying personalized dosing of immune suppressive agents or altering immune suppressive regimens. This may account for COVID-19 vaccination, but perhaps also for vaccinations against other diseases.

Our previous studies have demonstrated an association between kidney function and the humoral response after vaccination [4,5]. It is known that impaired kidney function alters the immune system due to premature aging and chronic systemic low-grade inflammation, resulting in a diminished response to vaccinations [25]. The antibody levels after COVID-19 vaccination are stepwise decreasing from healthy individuals to patients with chronic kidney disease to those receiving dialysis. This finding is confirmed in the current analysis, where kidney function remained significantly associated with antibody response after vaccination in the KTR longer after transplantation.

The main strengths of our study include its prospective design and the assessment of the humoral response on a continuous scale. Furthermore, the association between TTV and the humoral response was investigated after initial COVID-19 vaccination and sequential COVID-19 vaccinations. However, certain limitations should also be acknowledged. First, although a relatively large cohort of kidney transplant recipients was included, the results may not fully represent the entire population of KTR. Replication of our finding in an independent, larger cohort is needed to determine when the association between TTV load and antibody response changes over time after transplantation. Additionally, most patients were treated with low-dose prednisolone, tacrolimus, and MMF, which is the standard immunosuppressive regimen in many centers. We could, therefore, not study the relationship between immunosuppressive medication, TTV load, and the humoral response after vaccination for other immunosuppressive regimens.

In conclusion, in contrast to previous studies, we observed that the inverse association between TTV load and the humoral response following COVID-19 vaccination is not straightforward. The significant interaction between TTV load and time after transplantation indicates that TTV load predicts the humoral response, especially shortly after transplantation. Furthermore, 23% had undetectable TTV levels, and consequently, this marker could not be used in all patients. These findings emphasize the need for careful use and interpretation of TTV as a predictor for the humoral response after COVID-19 vaccination in KTR in clinical practice.

## Figures and Tables

**Figure 1 viruses-15-02189-f001:**
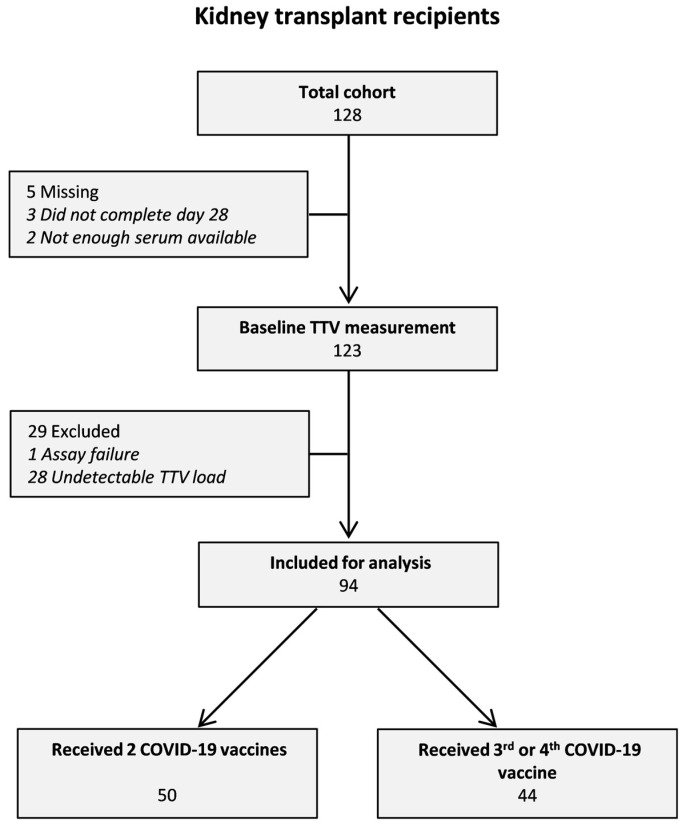
Subject enrollment.

**Figure 2 viruses-15-02189-f002:**
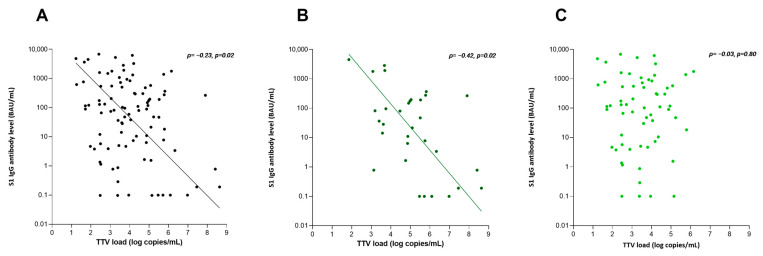
Scatter plot with Spearman correlation coefficient showing the association between TTV load at baseline and S1 IgG antibody level 28 days after the second vaccination. (**A**) The total cohort of KTR. (**B**) KTR before 24 months after transplantation. (**C**) KTR after 24 months after transplantation. The lines in (**A**,**B**) were calculated using orthogonal regression analysis.

**Table 1 viruses-15-02189-t001:** Baseline characteristics and kidney transplant recipients organized per TTV tertile.

	Total Cohort	TTV-Load		
		≤3.38	>3.38–<4.75	≥4.75
	(n = 94)	(n = 32)	(n = 31)	(n = 31)
Female, n (%)	41 (43.6)	14 (43.8)	14 (45.2)	13 (41.9)
Caucasian, n (%)	91 (96.8)	32 (100)	30 (96.8)	29 (93.5)
Age (years)	58.2 ± 12.2	58.5 ± 12.9	57.5 ± 11.0	58.5 ± 13.3
BMI (kg/m^2^)	27.2 ± 4.7	28.1 ± 5.1	26.9 ± 4.5	26.6 ± 4.5
Comorbidities, n (%)				
- Hypertension	70 (74.5)	20 (62.5)	23 (74.2)	27 (87.1)
- Diabetes mellitus	19 (20.2)	5 (15.6)	8 (25.8)	6 (19.4)
- History of malignancy ^1^	10 (10.6)	3 (9.4)	5 (16.1)	2 (6.5)
- Auto-immune disease	8 (8.5)	3 (9.4)	2 (6.5)	3 (9.7)
Lymphocytes (10^9^/L)	1.35 (0.9–2.0)	1.29 (0.8–1.7)	1.45 (0.9–2.1)	1.41 (0.9–2.0)
eGFR (mL/min/1.73 m^2^)	50.9 ± 18.9	46.5 ± 20.3	53.8 ± 17.8	52.6 ± 18.2
Primary renal diagnosis, n (%)				
- Immune-mediated disease	11 (11.7)	4 (12.5)	3 (9.7)	4 (12.9)
- Interstitial nephritis	7 (7.4)	6 (18.8)	1 (3.2)	0 (0.0)
- Familial/hereditary renal diseases	18 (19.1)	7 (21.9)	4 (12.9)	7 (22.6)
- Congenital diseases	9 (9.6)	4 (12.5)	3 (9.7)	2 (6.5)
- Vascular diseases	8 (8.5)	2 (6.3)	2 (6.5)	4 (12.9)
- Diabetic kidney disease	6 (6.4)	1 (3.1)	2 (6.5)	3 (9.7)
- Other	13 (13.8)	1 (3.1)	8 (25.8)	4 (12.9)
- Unknown	22 (23.4)	7 (21.9)	8 (25.8)	7 (22.6)
Transplant characteristics				
- First kidney transplant, n (%)	83 (88.3)	27 (84.4)	28 (90.3)	28 (90.3)
- Time after last transplantation (months)	41.0 (13.0–85.0)	63.5 (34.3–114.3)	45.0 (25.0–88.0)	13.0 (6.0–54.0) *
- Last transplant				
o Living, n (%)	62 (66.0)	24 (75.0)	20 (64.5)	18 (58.1)
o Pre-emptive, n (%)	39 (41.5)	14 (43.8)	13 (41.9)	12 (38.7)
Immunosuppressive treatment, n (%)				
- Steroids	93 (98.9)	31 (96.9)	31 (100)	31 (100)
- Mycophenolate mofetil	84 (89.4)	28 (87.5)	28 (90.3)	28 (90.3)
- Calcineurin inhibitor	89 (94.7)	28 (87.5)	31 (100)	30 (96.8)
- Azathioprine	3 (3.2)	2 (6.3)	0 (0.0)	1 (3.2)
- mTOR inhibitor	2 (2.1)	0 (0)	1 (3.2)	1 (3.2)
Tacrolimus trough level (µg/L)	5.0 (4.3–6.3)	4.7 (4.2–5.3)	4.9 (4.1–6.1)	6.2 (4.9–7.3) **
Number of received COVID-19 vaccinations				
- 2	50 (53.2)	14 (43.8)	16 (51.6)	20 (64.5)
- 3	33 (35.1)	15 (46.9)	11 (35.5)	7 (22.6)
- 4	11 (11.7)	3 (9.4)	4 (12.9)	4 (12.9)
Time between vaccinations (months)				
- Between 2nd and 3rd vaccination	6 (6–7)	-	-	-
- Between 3rd and 4th vaccination	3 (2–3)	-	-	-
S1 IgG antibody level after last COVID-19 vaccine (BAU/mL)	90.0 (5.2–527.8)	125.3 (5.4–942.0)	96.5 (14.1–841.0)	21.4 (0.8–196.4)

Variables are presented as mean ± SD, or as median (IQ interval) in case of non-normal distribution. *p*-values were calculated using one-way ANOVA in case of normal distribution, Kruskal–Wallis in case of non-normal distribution, and chi-square in case of proportion. Abbreviations are: BMI, body mass index; BAU, binding antibody units; eGFR, estimated glomerular filtration rate; TTV, Torque teno virus. ^1^ Including melanomas, excluding all other skin malignancies. * *p* < 0.001. ** *p* = 0.004.

**Table 2 viruses-15-02189-t002:** Associations of baseline characteristics with TTV load at baseline (n = 94).

	Univariable			Multivariable			Model 1		
	B (95% CI)	St. β	*p*-Value	B (95% CI)	St. β	*p*-Value	B (95% CI)	St. β	*p*-Value
Age (years)	−0.001 (−0.0–0.0)	−0.09	0.41						
Female sex	−0.01 (−0.09–0.06)	−0.04	0.71						
Time after transplantation (months)	−0.15 (−0.2–−0.1)	−0.44	<0.001	−0.16 (−0.2–−0.08)	−0.41	<0.001	−0.14 (−0.2–−0.1)	−0.42	<0.001
Tacrolimus trough level (µg/L)	0.61 (0.3–1.0)	0.39	<0.001	0.32 (−0.02–0.6)	0.20	0.06			
Hypertension (no vs. yes)	0.10 (0.02–0.18)	0.25	0.02	0.08 (−0.0–0.2)	0.19	0.06	0.1 (0.01–0.2)	0.21	0.03

Standardized beta and *p*-values were calculated using linear regression. Dependent variable is log-transformed TTV load. Time after transplantation and tacrolimus trough level were log-transformed. Abbreviations are: BMI, body mass index. Model 1: stepwise backward analysis taking out the least significant variable.

**Table 3 viruses-15-02189-t003:** Associations of baseline characteristics and TTV load with S1 IgG antibody level (BAU/mL) after vaccination (n = 94).

	Univariable			Multivariable			Model 1		
	B (95% CI)	St. β	*p*-Value	B (95% CI)	St. β	*p*-Value	B (95% CI)	St. β	*p*-Value
TTV load (log copies/mL)	−2.3 (−3.8–−0.8)	−0.30	0.004	−1.55 (−3.2–0.1)	−0.21	0.07	−2.3 (−3.7–−1.0)	−0.30	<0.001
Hypertension (no vs. yes)	−0.71 (−1.3–−0.08)	−0.20	0.03	−0.22 (−0.8–0.4)	−0.07	0.48			
eGFR (mL/min/1.73 m^2^)	0.03 (0.1–0.4)	0.39	<0.001	0.03 (0.02–0.04)	0.43	<0.001	0.03 (0.02–0.05)	0.45	<0.001
Donor type of last transplant (living vs. deceased donor)	−0.6 (−1.2–−0.02)	−0.21	0.04	−0.42 (−1.0–0.1)	−0.15	0.12		-	
Time after transplantation (months)	0.62 (0.1–1.1)	0.24	0.02	0.44 (−0.2–1.1)	0.15	0.17			
MMF use (no vs. yes)	−0.77 (−1.7–0.1)	−0.17	0.09	-0.78 (−1.7–0.2)	−0.17	0.10	−1.2 (−2.0–−0.4)	−0.26	0.004
Tacrolimus trough level (µg/L)	−3.7 (−6.3–-1.2)	−0.33	0.004	−1.55 (−40–0.9)	−0.14	0.21			
Number of received COVID-19 vaccinations (2 vs. 3 or 4 vaccinations)	0.60 (0.04–1.1)	0.22	0.04	0.53 (−0.03–1.1)	0.20	0.06	0.67 (0.2–1.2)	0.25	0.008

Standardized beta and *p*-values were calculated using linear regression. Dependent variable is log-transformed SARS-CoV-2 S1 IgG antibody levels. Time after transplantation, TTV load, and tacrolimus trough level are log-transformed. Abbreviations are: eGFR, estimated glomerular filtration rate; MMF, mycophenolate mofetil; TTV, Torque teno virus; Model 1: stepwise backward analysis taking out the least significant variable.

**Table 4 viruses-15-02189-t004:** Associations of TTV load and time after transplantation with S1 IgG antibody level (BAU/mL) after vaccination stratified for patients ≤ 24 months (n = 32) or > 24 months (n = 62) after transplantation.

	Time after Transplantation ≤ 24 Months						Time after Transplantation > 24 Months					
	Univariable			Model 1			Univariable			Model 1		
	B (95% CI)	St. β	*p*-Value	B (95% CI)	St. β	*p*-Value	B (95% CI)	St. β	*p*-Value	B (95% CI)	St. β	*p*-Value
TTV load (log copies/mL)	−5.3 (−8.3–−2.3)	−0.55	0.001	−5.3 (−8.3–−2.3)	−0.55	0.001	−0.61 (−2.7–1.5)	−0.08	0.55			
Time after transplantation (months)	2.08 (0.5–3.7)	0.44	0.01				−0.15 (−1.3–1.0)	−0.03	0.79			
BMI (kg/m^2^)	0.10 (−0.01–0.2)	0.31	0.08									
MMF use (no vs. yes)	−1.61 (−3.3–0.1)	−0.34	0.06									
Tacrolimus trough level (µg/L)	−5.6 (−10–−0.1)	−0.44	0.02									
Donor type of last transplant (living vs. deceased donor)							−0.64 (−1.3–0.7)	−0.23	0.07	−0.62 (−1.2–−0.03)	−0.22	0.04
eGFR (mL/min/1.73 m^2^)							0.03 (0.2–0.5)	0.51	<0.001	0.03 (0.02–0.05)	0.51	<0.001
Number of received COVID-19 vaccinations (2 vs. 3 or 4 vaccinations)							0.62 (−0.03–1.3)	0.24	0.06	0.57 (0.03–1.1)	0.22	0.04

Standardized beta and *p*-values were calculated using linear regression. Dependent variable is log-transformed SARS-CoV-2 S1 IgG antibody level. Time after transplantation, TTV load, and tacrolimus trough level are log-transformed. Abbreviations are: BMI, body mass index; eGFR, estimated glomerular filtration rate; MMF, mycophenolate mofetil; TTV, Torque teno virus; Model 1: stepwise backward analysis taking out the least significant variable.

## Data Availability

The data presented in this study are available on request from the corresponding author.

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
