# Peer review of "The Time-Dependent Association of Torque Teno Virus Load with the Level of SARS-CoV-2 S1 IgG Antibodies Following COVID-19 Vaccination in Kidney Transplant Recipients"

_viruses, 2023, doi:10.3390/v15112189_

Round 1
Reviewer 1 Report
Comments and Suggestions for Authors
Céline Imhof and colleagues reported an investigation on the time-dependent association of Torquetenovirus (TTV) load with the level of SARS-CoV-2 S1 IgG antibodies following COVID-19 vaccination in Kidney Transplant Recipients. Interestingly, they found a significant association between TTV load and levels of S1 antibody after short time of transplantation but not at longer time. The authors claimed that these finding suggest TTV load as potential good marker to predict COVID-19 vaccination antibody response and helpful to select a vaccination strategy shortly after transplantation. The study reports interesting data and confirm TTV as potential marker of immune functionality. However, there are some points that need additional clarification.
Main points
1- the most relevant information that need to be improved is the row data of antibody titer used in the statistical elaboration of the data. For the reader is hard to understand the data in the tables without the clear indication of the titer of the antibody anti-S1 used. Moreover, the indication of “COVID-19 vaccines (2 vs 3 or 4)” is not clear. For examples, for TTV is reported (log copies/ml). it is possible add the indication also for antibody?
2- The authors should report if the association of TTV load and S1 antibody titer have a correlation positive or an inverse correlation. From the figure 2 the latter is. If it is right, this should be commented better in the discussion.
3- A brief description of the method used to test antibody anti S1 should be added in section 2.2.
4- finally, the authors have to make clear the meaning of “TTV load helpful to select vaccination strategies”. What type of clinical decision could have the information of TTV load in the selection of vaccine?
Minor points
1- Torque tenovirus or Torqe Teno virus is reported alternatively in the text (line 41 and 60). The use of only one type of name should be appropriate.
2- line 36: What is the meaning of MMF ?
3- All typos should be corrected.
Reviewer 2 Report
Comments and Suggestions for Authors
Céline Imhof and co-authors wrote an interesting and timely work, and the TTV virus is a novel marker of the immune status of patients after organ transplantation. Vaccination is the best way to protect patients against infectious diseases. Patients undergoing immunosuppressive treatment respond poorly to vaccinations, and monitoring antibody levels after vaccination is not a routine practice. The manuscript has a practical impact on the daily practice of a transplantologist. The question remains how to monitor the immunological status of patients late after organ transplantation.
Good luck !
Reviewer 3 Report
Comments and Suggestions for Authors
Congratulations to the authors for a very good, clinically important work. I recommend publishing the manuscript in the version sent to the editor.
Reviewer 4 Report
Comments and Suggestions for Authors
In this manuscript, Imhof and al. expand on previous observations of the predictive potential of TTV viral load for vaccine response by analyzing the humoral response as a continuous variable.
The cohort is rather small compared to some previous studies (92 patients actually included in the analysis) and lacks detailed information on several points. Further analyses should also be done.
The major problem is the exclusion of 25% of the patients, not in a random way (e.g. missing samples) but on the basis on a low TTV load, which creates a major bias. As TTV viral load was only investigated on the baseline sample timepoint, a negative result does not exclude TTV infection. Indeed, TTV prevalence in the population is rather >95% so the majority of these negative patients may actually harbor TTV, but have a low/negative viral load at this timepoint (reflecting a strong immune system). Before excluding these patients, TTV viral load should be measured on the other timepoints: samples 28 days after the second, third or fourth dose. Patients could be excluded only if they are negative on all samples with at least 3 negative timepoints.
Other comments:
- Were patients naïve from COVID infection before vaccination?
- What is the interval between the second, third and fourth doses ?
- on figure 1, TTV measurement timepoint is not mentioned
- additionally to my major comment, a 1 in 2 dilution was used prior to extraction, lowering the chance of detecting low TTV viral loads
- table 1, why are tertiles used and not other thresholds ?
- according to the response this remark could also be a major comment: in tables 3 and 4, at which timepoint are the IgG antibody levels compared? Are all timepoints grouped? Antibody levels should be compared at the same timepoint/number of doses otherwise it may be another bias since not all patients received a third or fourth dose. Additionnally, the number of patients having received a third or fourth dose should be mentioned; I do not really understand the "number of previous COVID vaccines" as 0, 2 or 3 in the tables.
- lastly, while this observation of the performances of TTV as a continuous variable as well as antibody levels is intellectually interesting, the use in clinical practice should be mentioned. Line 214, what does "striving for an as high antibody level" mean? Revaccinate the patient as frequently as possible (regardless of TTV load) to reach titers above measurement limits? For TTV, how do authors suggest the use of the viral load as a continuous variable?
- I believe references 20-22 are not mentioned in the text.
Comments on the Quality of English LanguageModerate editing of English language required. "Therefore" is mentioned in the second and third sentences of the abstract. The following sentence starts with "122 were included KTR". I also suggest finding another wording for "patients were longer after transplantation".
Author Response
"Please see the attachment."

Round 2
Reviewer 1 Report
Comments and Suggestions for Authors
The authors have substantially improved the manuscript. Now the study is suitable for pubblication.
Reviewer 4 Report
Comments and Suggestions for Authors
All comments were adequately answered.